# Formation of Polymer-Carbon Nanotube Composites by Two-Step Supercritical Fluid Treatment

**DOI:** 10.3390/ma14237428

**Published:** 2021-12-03

**Authors:** Anton M. Vorobei, Konstantin B. Ustinovich, Sergei A. Chernyak, Sergei V. Savilov, Olga O. Parenago, Mikhail G. Kiselev

**Affiliations:** 1Kurnakov Institute of General and Inorganic Chemistry of the Russian Academy of Sciences, 119071 Moscow, Russia; ystinovich@supercritical.ru (K.B.U.); savilov@mail.ru (S.V.S.); oparenago@scf-tp.ru (O.O.P.); 2Chemical Department, Lomonosov Moscow State University, 119991 Moscow, Russia; madseryi@mail.ru; 3G. A. Krestov Institute of Solution Chemistry of the Russian Academy of Sciences, 153045 Ivanovo, Russia; mgk@isc-ras.ru

**Keywords:** CNT-polymer composites, supercritical antisolvent precipitation (SAS), rapid expansion of supercritical suspensions (RESS)

## Abstract

An approach for polymer-carbon nanotube (CNT) composite preparation is proposed based on a two-step supercritical fluid treatment. The first step, rapid expansion of a suspension (RESS) of CNTs in supercritical carbon dioxide, is used to de-bundle CNTs in order to simplify their mixing with polymer in solution. The ability of RESS pre-treatment to de-bundle CNTs and to cause significant bulk volume expansion is demonstrated. The second step is the formation of polymer-CNT composite from solution via supercritical antisolvent (SAS) precipitation. SAS treatment allows avoiding CNT agglomeration during transition from a solution into solid state due to the high speed of phase transition. The combination of these two supercritical fluid methods allowed obtaining a polycarbonate-multiwalled carbon nanotube composite with tensile strength two times higher compared to the initial polymer and enhanced elasticity.

## 1. Introduction

CNT-polymer composites have attracted a lot of attention in the last two decades. The main reason for this is the possibility to design materials with unique mechanical, electrical and thermal properties. CNT-polymer composites have great prospects for many potential applications. Their excellent electrochemical charge storage properties and fast charge/discharge switching make it possible to use these composites in high power supercapacitors [1,2,3,4]. CNT-polymer materials can be used as strain sensors [5,6,7,8] due to the linear dependence of electrical resistance on reversible mechanical deformations. Other potential applications include photovoltaic devices [9,10], laser equipment [11], nano-electronics [12,13], gas sensors [14,15], membranes [16,17], etc.

However, there are some obstacles to full-scale implementation of polymer–CNT composite materials in industrial practice. The main problem in the preparation of these composites is agglomeration of CNTs [18,19,20]. Conventional methods such as melt blending, solution processing and in situ polymerization typically work with a liquid phase whereas the final product is a solid. The transition from the liquid to the solid phase often takes a long time which gives rise to CNT agglomeration. Composites prepared by conventional techniques often have inhomogeneous distribution of CNTs in the polymer matrix. This leads to deterioration of their mechanical and electrical properties [19]. Kashiwagi et al. [20] demonstrated that the storage modulus, electrical conductivity, and flammability property of the nanocomposites correlate well with the relative dispersion index representing the uniformity of the dispersion of CNTs. For the nanocomposites containing the same amount of CNTs, the relationships between the quantified dispersion levels and physical properties show about four orders of magnitude variation in storage modulus and almost eight orders of magnitude variation in electric conductivity. In order to reduce the agglomeration of CNTs in the process of composite preparation it is necessary to minimize the time of its transfer from solution into a solid material.

Another challenge of CNT-polymer composites formation is the problem of obtaining a well-dispersed suspension of CNTs in a solvent. The most commonly used and effective method for CNT dispersion is ultrasonication, especially, a high-frequency ultrasonication mixing technique along with the axial flow impeller called ultrasonic dual mixing (UDM) technique. Recently, Rathi and Kundalwal [21] obtained multi-walled CNTs/ZrO_2_-based hybrid epoxy nanocomposites using this method. The fracture toughness of such composites was improved by ~31% compared to the neat epoxy when 1.0 wt% loading of CNT/ZrO_2_ hybrid composite nanofillers was used to fabricate MNC Unfortunately, powerful ultrasound treatment can also affect CNT length and diameter [22,23,24]. F. Hennrich et al. demonstrated via atomic force microscopy that after 40 min of ultrasound treatment, 800 ± 300 nm long single-walled carbon nanotubes have been cut down to a length of ∼200 nm [22]. Lucas et al. [23] measured the average length L(t) of the multi-walled nanotubes as a function of ultrasound treatment time t using dynamic light scattering, and observed that L(t) scales as t^−n^, with n ≅ 0.2. Moreover, the ultrasonication can produce various defects in the CNT structure [25,26] or promote formation of super-ropes. The latter can be more than 20 times larger in diameter than the initial bundle [27]. Ultrasonication of MWCNTs results in expansion and peeling or fractionation of MWCNT graphene layers [28]. Consequently, an urgent problem now is the search for alternative non-destructive methods of CNT dispersion.

Supercritical fluids as media for materials processing have a number of advantages over liquid solvents, for example, low viscosity, high diffusivity, absence of surface tension, etc. [29]. Supercritical fluids can be effectively used in processes which require a high speed of liquid-to-solid transition. The most commonly used substance in supercritical fluid technologies (SCFT) is carbon dioxide because it is cheap, environmentally friendly, reasonably inert chemically and gaseous at atmospheric pressure and room temperature. Evaporation of carbon dioxide after process completion is achieved automatically by a pressure release. In contrast, evaporation of liquid solvents is usually costly, time-consuming and sometimes environmentally hazardous [20,21,22,23,24,25,26,27,28,29,30,31,32].

The aim of this work was twofold—to test the efficiency of the new method of debundling CNTs by the so-called rapid expansion of supercritical suspensions (RESS) and to form a composite material from a polymer and RESS-debundled CNTs by supercritical antisolvent (SAS) precipitation assessing the joint effect of the two consecutive supercritical fluid treatments on the composite mechanical characteristics.

The main idea underlying the RESS method is the ability of a supercritical fluid to expand rapidly at a pressure release with a complete loss of its solvating power. A solid material is dissolved or dispersed in a supercritical fluid, most frequently carbon dioxide. Then, rapid spraying of the solution through a heated nozzle into a low pressure chamber leads to an abrupt loss of solvating power and, hence, rapid nucleation of the substrate in the form of small particles that are collected from the gaseous stream [30]. The use of a suspension modification of RESS for nanomaterial pretreatment was reported several times [33,34,35]. For instance, in [35] RESS was used for effective deagglomeration of suspensions of alumina and titania nanopowders: the number weighted mode diameters measured by the scanning mobility particle spectrometer were below 100 nm in all cases. In this work, we apply rapid expansion of a CNT suspension in SC-CO_2_ for debundling nanotubes prior to mixing them with a polymer. The main incentive for using this approach is that during RESS CNT bundles are expected to disentangle due to the high kinetic energy of the expanding fluid. The resulting expanded material must be more susceptible to uniform mixing with a polymer solution.

Previously we have shown the applicability of the supercritical antisolvent (SAS) precipitation method to the formation of polymer-CNT composites [36]. The main idea of this process is the same as in liquid antisolvent precipitation. A liquid solution of target compounds is sprayed through a nozzle into a constant flow of a supercritical fluid antisolvent. The supercritical fluid must be completely miscible with the liquid solvent, whereas the target solutes must be insoluble in the SCF. The liquid solution contact with the SCF induces supersaturation and rapid precipitation of the solute [37]. Subsequent flushing of the precipitation chamber with an additional portion of the SCF allows rinsing the precipitated product from the traces of organic solvents. SAS is widely used for micronization of pharmaceutical substances [37,38,39,40,41], biocompatible polymers [42,43,44], formation of catalysts [45,46], superconductor precursor nanoparticles [47], explosives and high propellants [48], etc. The main advantage of the SAS process for preparation of CNT-polymer composites is very fast solute precipitation. As a result, CNTs have very short time for agglomeration and it is possible to achieve homogeneous distribution of CNTs in the polymer matrix.

## 2. Materials and Methods

### 2.1. Materials

Multiwalled CNTs were synthesized by catalytic chemical vapor deposition method using hexane as a carbon precursor in the presence of Co−Mo/MgO catalyst. For this purpose, the catalyst, obtained by the combustion method from Mg(NO_3_)_2_·6H_2_O, Co(NO_3_)_2_·6H_2_O, (NH_4_)_2_Mo_2_O_7_, citric acid and glycine (Chimmed, Moscow, Russia), was placed inside a quartz tubular reactor. The reactor then was heated to 750 °C in N_2_ flow. Then, N_2_ flow was switched to bubble through hexane for 5 h. The obtained material was cooled in N_2_ flow up to 400 °C and then cooled to room temperature in air atmosphere to remove amorphous impurities. The resultant powder was refluxed in concentrated HCl for 3 h to dissolve the catalyst. Finally, material was filtered and washed in deionized water until neutral pH and dried at 130 °C for 12 h [49]. The CNT carboxylation was performed through refluxing in concentrated HNO_3_ for 6 h with subsequent rinsing and drying. The oxygen mass content in the carboxylated CNTs was 10.9%. The CNT mean diameter was 15–30 nm.

The polycarbonate (PC) Wonderlite^®^ PC-110U was purchased from “Rusplast” OOO Chimmed (Moscow, Russia). The chloroform 99.9% was from Chimmed (Moscow, Russia). The food grade carbon dioxide (99.8%) and nitrogen was from LindeGasRus (Balashikha, Russia).

### 2.2. Rapid Expansion of Supercritical Suspensions (RESS) Carbon Nanotube (CNT) Debundling

Waters (Pittsburgh, PA, USA) RESS-100 equipment was used for CNT debundling. Its scheme is shown in Figure 1.

We placed 500 mg of cylindrical carboxylated CNTs into a high pressure vessel (5). The vessel was sealed and the CO_2_ cylinder was opened to fill the system. Then the pressure was increased up to 200 bar and the CNTs were refluxed in supercritical carbon dioxide for 15 min at 40 revolutions per minute. The vessel temperature was maintained at 40 °C. After that the suspension was sprayed into the precipitation chamber (7) through a heated 100 µm nozzle by manually opening valve (6). The precipitation chamber was held at atmospheric pressure and ambient temperature. This procedure was repeated three times to achieve full transfer of the carbon nanotubes from the vessel (5) to the precipitation chamber (7).

### 2.3. Preparation of Polymer-CNT Solutions

A sample of PC was dissolved in 240 mL of chloroform. An ultrasonic bath was used to increase the polymer dissolution rate. Then a weighted sample of CNTs was added to the polymer solution. The PC concentration in the solution was 50 g/L, unless stated otherwise, the CNT/polymer mass ratio was maintained at 0.6% in all the experiments. The solution ultrasonication was performed either using an ultrasonic bath (100 W, Quick 218–100) for 4–5 h to obtain a stable CNT suspension or through titanium ultrasonic horn processing at 600 W for 20 min. In the case of ultrasonic horn processing, ice was used to prevent suspension overheating.

### 2.4. Supercritical Antisolvent (SAS) Composite Precipitation

The Waters’ SAS-50 system was used for supercritical antisolvent precipitation, and its scheme is represented in Figure 2. The SAS experiments were carried out as follows. The solution pump (7) was initially filled with pure solvent and then connected to a reservoir containing CNT-polymer suspension. CO_2_ pump (4), automatic back pressure regulator (ABPR) 9 and all the heat exchangers were started and controlled via Process Suite software (Pittsburgh, PA, USA). Upon reaching the operating temperature, pressure and CO_2_ flow rate, the CNT suspension in the polymer solution was sprayed by pump (7) through a spraying nozzle into high-pressure precipitation vessel (8). The suspension rapidly mixed with SC-CO_2_, which led to an abrupt drop of the solvating power and formation of a highly supersaturated solution. Fine PC-CNT composite particles formed and precipitated in vessel (8), whereas the CO_2_-solvent mixture flows through it. After completing the sample spraying, additional 20–25 mL of pure solvent were pumped through the system to wash all the tubes and blocks before the spraying nozzle. To wash the precipitated composite, the CO_2_ flow was maintained for the time required to pump two volumes of vessel (8). Then the CO_2_ flow is stopped and depressurization is carried out using ABPR (9). The composite powder is extricated from vessel (8) using a built-in basket.

Prior to composite precipitation, the SAS process of pure PC solution was optimized in the same manner as in the case of composite precipitation, but without adding CNTs. The optimization experiments were carried out in the following working conditions: a temperature of 40 °C, CO_2_ flow rate of 50 g/min, pressure of 150 bar and 200 bar, solution flow rate of 1 and 2 mL/min, nozzle diameter of 100 µm, PC concentration in the solution of 25 and 50 g/L.

The CNT-polymer composites were obtained at a temperature of 40 °C, solution flow rate of 1 mL/min, CO_2_ flow rate of 50 mL/min, pressure of 150 bar, nozzle diameter of 100 µm, and PC concentration of 25 g/L. Chloroform was used as the solvent in all the experiments.

### 2.5. Preparation of Composites by the Solution Processing Method for Comparison

For comparison of the composite mechanical properties some samples were prepared by the conventional solution processing method. One gram of PC was dissolved in 40 mL of chloroform. We added 6 mg of CNTs to the obtained solution. The mixture was ultrasonicated for 4–5 h. The resulting suspension was poured into a Petri dish and left to dry at room temperature. The composite film obtained was cut into small pieces. These pieces were used to form samples for the measurement of mechanical properties by the hot pressing method.

### 2.6. Preparation of Samples for Measuring Mechanical Properties by the Hot Pressing Method

The composites obtained by either SAS or solution processing methods were used to form samples for measuring mechanical properties. A silicone sheet with a shovel-like cavity was used as a mold. The composite powder to be pressed was placed into this cavity. The powder mass was 1 g. Then the silicone sheet was covered on two sides with aluminum foil. The pressing temperature was 200 °C. The materials were kept under pressure for 15 min at the pressing temperature and additionally for 10 min required for the press to cool.

### 2.7. Scanning Electron Microscopy

The electron micrographs of samples surfaces were obtained with a scanning electron microscope «JEOL JSM-6390LA» (JEOL Ltd., Tokyo, Japan). The sample was put onto a carbon conductive bilateral adhesive tape pasted on a copper-zinc table. The samples were covered with a 2.5 nm thick layer of gold by the magnetron sputtering method. This procedure was performed using Quorum Q150R ES in vacuum. The accelerating voltage (from 0.5 to 30 kV), as well as the working distance (8–25 mm) was chosen depending on the sample structure. Some additional images were obtained with a Carl Zeiss NVision 40 scanning electron microscope. The accelerating voltage was set at 1 keV.

### 2.8. X-ray Photoelectron Spectroscopy

The X-ray photoelectron spectroscopy (XPS) spectra were acquired on an Axis Ultra DLD spectrometer (Kratos Analytical, Stretford, UK) using a monochromatic Al*K*_a_ source (hv = 1486.7 eV, 150 W). The pass energies of the analyser were 160 eV for survey spectra and 40 eV for high-resolution scans. The binding energy scale of the spectrometer was preliminarily calibrated using the position of the peaks for the Au 4f_7/2_ (83.96 eV), Ag 3*d*_5/2_ (368.21 eV) and Cu 2p_3/2_ (932.62 eV) core levels of pure metallic gold, silver, and copper. The powder CNT samples were fixed on a holder using a double-side conductive adhesive tape.

### 2.9. Transmission Electron Microscopy

To study the final composite by transmission electron microscopy (TEM), we prepared slices of the composite samples using ultramicrotomy. An ultramicrotome Reichert-Jung (Germany) was used for this purpose. The slice thickness was about 100 nm. The TEM study was carried out using JEOL JEM 2100 F/Cs equipped with an electron energy loss spectroscopy (EELS) analyzer and a chromatic aberration corrector (Cs).

### 2.10. Measurement of Mechanical Characteristics

The mechanical characteristics of the hot-pressed samples were investigated using a tensile testing machine TTM-5 (Trilogica, Reichelsheim, Germany). The thickness and width of each sample was measured using a micrometer before testing the mechanical characteristics. The measurements for each point were repeated at least three times.

## 3. Results and Discussion

### 3.1. Optimization of the SAS Process for Polycarbonate (PC)

Powder formation by SAS can be carried out successfully only under a specific set of conditions. Several scenarios should be avoided in order to obtain the desired result. First, if the solution jet does not break effectively enough after the nozzle, so-called «icicles» can be formed instead of a fine powder. Second, the micronized polymer can swell in the SC-CO_2_ medium and plug the filter at the bottom of the precipitation vessel. Third, if the polymer solubility in the CO_2_-solvent mixture is non-negligible, it can lead to substantial mass losses. Hereafter, we will term the experiments in which we succeeded in obtaining a polymer powder with the collection yield over 90% as “successful”. By «optimization» we mean selection of parameters, which lead to successful powder precipitation with the maximum possible yield.

Solution flow rate and polymer concentration in the solution are two process parameters having the most drastic effect on polycarbonate SAS. It was found that the solution flow rate should be maintained at a low value. At 1 mL/min, we achieve a successful process with a high yield (Figure 3a), whereas at 2 mL/min, «icicles» (macroscopic polymer growths) are mostly formed (Figure 3b). The formation of «icicles» leads to the spraying nozzle obstruction which blocks the system.

The polymer concentration in the solution should be also kept below a certain limit. Powder formation was observed, if the PC concentration in the solution was 25 g/L, whereas the increase to 50 g/L changed the precipitation mechanism and led to the formation of “icicles”, possibly due to higher solution viscosity.

Solution flow rate and polymer concentration do not only affect the successfulness of the SAS process per se, but also the morphology of the particles obtained. The SEM images of the PC particles after SAS micronization in cases of 1 mL/min and 2 mL/min solution flow rates are demonstrated in Figure 4a,b, respectively. At 1 mL/min, spherical particles of 200–2000 nm in diameter are obtained. Such morphology is preferable for future composite formation because it is more likely to allow maintaining the uniform distribution of the nano-additive within the polymer matrix. At 2 mL/min, irregular anisotropic structures are formed.

The further experiments for CNT-PC composite precipitation were carried out at a pressure of 150 bar, temperature of 40 °C, solution flow rates of 1 mL/min, CO_2_ flow rate of 50 g/min, and polymer concentration in the solution of 25 g/L.

### 3.2. Preparation of CNT-Polycarbonate Composite Powders by the SAS Method

The PC-CNT composite powder was obtained in the chosen conditions. Figure 5 shows a typical SEM image of the obtained composite. It indicates that the polymer covers the carbon nanotubes. In our opinion, such composite structure gives indirect evidence of the fact that nanotubes are not able to agglomerate after a SAS process.

### 3.3. Rapid Expansion of a CNT Suspension in Supercritical Carbon Dioxide

RESS pre-processing of CNTs resulted in a significant decrease in the CNT bulk density. The volumes of the same mass of CNT before and after RESS are presented in Figure 6. Moreover, RESS processing leads to certain changes in the structure of CNT bundles. The SEM images of the initial CNTs and CNTs after RESS are shown in Figure 7. It can be seen that the degree of CNT agglomeration significantly decreases after RESS. The mean size of the processed CNT agglomerates is about 7 µm.

Carbon and oxygen lines are observed in the XPS spectra of the samples (Figure 8). The high-resolution spectra of the oxygen and carbon lines for both samples are similar in shape (Figure 9a,b). These spectra were decomposed into components corresponding to different states of carbon and oxygen atoms. A previously obtained spectrum of well-purified multi-walled CNTs was used as a component corresponding to sp^2^ carbon. The results of the decomposition of the spectra into components are shown in Figure 10a,b and in Table 1. The attribution of components to certain states of atoms is carried out on the basis of data [50]. According to the results of the study, the differences between the samples are minimal. A small difference in the oxygen content on the surface of the samples cannot be considered significant, based on the typical accuracy of the results of the XPS studies.

### 3.4. Mechanical Properties of Composite Materials

The benefits of using SAS for CNT-polymer composite formation had been demonstrated previously [36]. In this work, we focused on the influence of different approaches to CNT processing prior to composite formation on the mechanical properties of the final composite. Table 2 summarizes the mechanical characteristics of the PC-CNT composites obtained by different techniques. In all cases introduction of CNTs into polymer matrix leads not only to increase of tensile strength, but also an increase of elongation of composites compared to initial PC. By incorporation of the nanofillers into the polymeric matrix, stiffness usually increases and elongation reduces. However, for CNTs the same observation (increase of strength and elongation) was reported in the case of different polymers, for example, poly(vinyl chloride) [51] and poly(vinyl butyral) [52]. This can be attributed to intrinsic flexibility and the high aspect ratio of CNTs at the same time.

#### 3.4.1. Effect of RESS Treatment

It can be concluded from the data of Table 2 that pre-processing of initial CNTs by RESS leads to additional improvement of the mechanical properties of the obtained composites. The most significant improvement is observed for the samples obtained by SAS (experiments No. 1, 3, 5, 6). The use of CNTs de-bundled by RESS (experiment No. 3) in the preparation of composites by SAS allows increasing the tensile strength of the final material by 20%. The total elongation of such composites (experiment No. 3) is also approximately 25% higher than in the composites formed by SAS without RESS pre-treatment (experiment No. 1). The overall increase in the tensile strength and total elongation achieved by the CNT introduction into the PC matrix is more than 2 and 3.8 times in case of the RESS-SAS two-step treatment. We believe that this is associated with the RESS ability to decrease the size of CNT bundles, as shown above (Figure 7b). Smaller agglomerates must result in higher homogeneity in the CNT suspension and in dispersion within the final material respectively. The similar results of RESS treated CNTs ultrasonication were demonstrated in [53] using ultraviolet–visible spectroscopy. Compared to untreated nanotubes, RESS processed CNTs were easily dispersed in an aqueous 0.5% sodium-dodecyl-sulfate solution. Such explanation is in good agreement with the TEM images (Figure 11). In both cases there are not large agglomerates of CNTs. However, the distribution of CNTs in the case of a composite with RESS-processed CNTs (experiment No. 3) is more homogeneous (Figure 11c). This is especially evident in Figure 11c compared to 11a. According to our information, only a few attempts to obtain polymer composite with RESS-processed CNTs are known. Chen et al. [54] used such a treatment before melt-blending process for CNTs-poly (phenylsulfone) composites preparation. That work also demonstrated improved CNT dispersion in the polymer matrix and more uniform networks formed in the case of RESS-processed CNTs. Such an increase of homogeneity of composite leads to the improvement of the strength and elasticity of modified samples compared to those without additional treatment.

Introduction of RESS-treated CNTs into polycarbonate by the solution processing method also leads to an increase in tensile strength and total elongation growth, albeit to a lesser extent. The increase in the tensile strength and total elongation is by 14 and 30%, respectively in the case of experiment No. 4, compared with the composites that contain untreated CNTs (experiment No. 2). However, the solution processing method does not allow us to achieve the results of the SAS method alone (experiment No. 1) even if it uses RESS-processed CNTs (experiment No. 4). The tensile strength and total elongation of composites obtained by SAS is 20% and 50% higher respectively compared to those obtained via solution processing with RESS-processed CNTs. In the TEM images (Figure 12) we can see more large CNT aggregates, especially in experiment 2, than in the case of SAS samples (Figure 11). Thus, we attribute this to the fact that in the solution processing method used in this work, the CNTs have enough time for agglomeration and bundling during the transition from solid to liquid phase. Solvent evaporation takes a long time, during which the CNT de-bundling achieved by RESS can partially deteriorate.

#### 3.4.2. Effect of Powerful Ultrasound Processing

The use of powerful ultrasound (experiment No. 5) virtually does not change the composite tensile strength, but leads to a two-fold increase in the total elongation in comparison with a regular ultrasonic bath (experiment No. 1). On the one hand, CNTs distribution in a polymer matrix for composites obtained using powerful ultrasound should be more homogeneous than those in the case of ultrasonic bath. Therefore, the ultrasonic horn should lead to improvement of composite elasticity and strength. On the other hand, it can significantly reduce CNTs length and produce defects. Moreover, it can affect other CNT characteristics, such as waviness, which can be critical for elasticity. In fact, large CNTs agglomerates in TEM images (Figure 13a) were not observed in the case of the composite (experiment No. 5) and its CNTs distribution is more homogeneous than in the case of experiment No. 1. However, it is hard to determine definitely using TEM images is the tubes length significantly different in these cases.

Unexpected results are obtained when powerful ultrasound and RESS pre-processing are combined (experiment No. 6). The improvement of mechanical characteristics in this case is the least of all the SAS experiments. It is still higher than the solution processing values, but even SAS alone without any additional treatment (experiment No. 1) gives bigger growth in the tensile strength and elongations. Meanwhile, the homogeneity of CNTs distribution in sample 6 (Figure 13b) is not significantly lower compared to sample 1. The reason for such behavior is unclear at the moment. One possible explanation could be built on the assumption that RESS-treated CNTs are more fragile and more prone to destruction by ultrasound. CNT length is one of the factors influencing the mechanical properties. Typically, longer CNTs result in higher strengths and elongation. If RESS-treated CNTs break and become shorter when subjected to powerful ultrasound, their impact on the mechanical strength deteriorates even if the dispersion homogeneity does not change. That could explain smaller enhancements in this case. Another possible explanation is that RESS treatment does not only de-bundle CNTs but also changes the bundle nanostructure which might affect the toughening mechanism inherent in them [55]. One can speculate that RESS can change the overall waviness of CNTs or make the bundle surface less smooth. This could lead to an increase in the crack deflection ability [55]. If the RESS-caused bundle structure changes become less pronounced under powerful ultrasound, this fact might explain the diminished effect of CNT introduction in case of double pre-treatment. Additional research is required to find out the exact reason for the observed phenomenon.

## 4. Conclusions

Treatment of bundled CVD-synthesized multi-wall CNTs by the RESS method leads to significant bulk expansion and effective de-bundling, which is beneficial for their further application as additives for the construction of composite materials. By combining RESS pre-treatment and SAS precipitation, it is possible to obtain polymer-CNT composites whose tensile strength is more than two times higher than that of the initial polymer. The elastic properties of material obtained in such a manner are also enhanced. Powerful ultrasound treatment of a CNT suspension in a polymer solution prior to SAS is the most effective approach to improving the composite elasticity.

## Figures and Tables

**Figure 1 materials-14-07428-f001:**
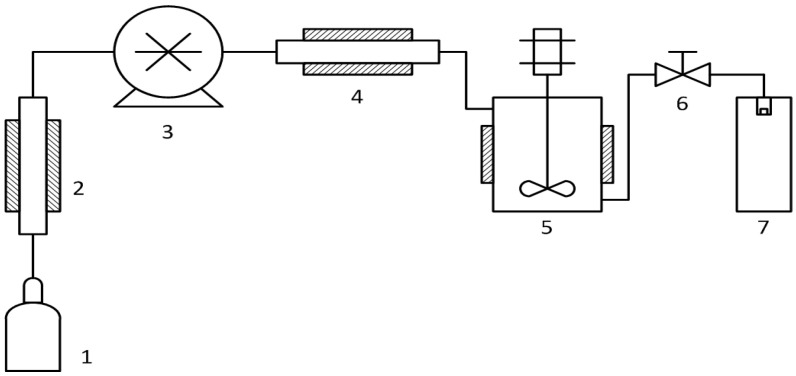
Schematic representation of the rapid expansion of supercritical suspensions (RESS) experimental setup. 1 is a CO_2_ cylinder; 2 is a cooler; 3 is a CO_2_ pump; 4 is a heater; 5 is a high pressure vessel with a magnetically driven 4-blade stirrer; 6 is a valve; 7 is a precipitation chamber with a heated spraying nozzle.

**Figure 2 materials-14-07428-f002:**
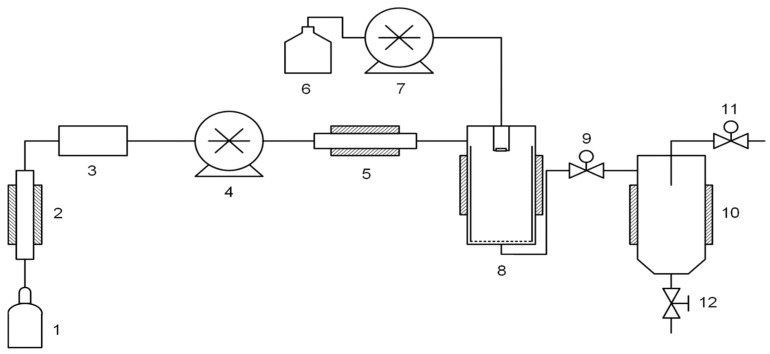
Schematic representation of the supercritical antisolvent (SAS) apparatus adapted from [36]. *1* is a CO_2_ container; *2* is a cooler; *3* is a flowmeter; *4* is a CO_2_ pump; *5* is a heater; *6* is a polymer solution; *7* is a solution pump; *8* is a precipitator; *9* is an automatic back pressure regulator; *10* is a separator; *11* is a manual back pressure regulator; *12* is a drain valve.

**Figure 3 materials-14-07428-f003:**
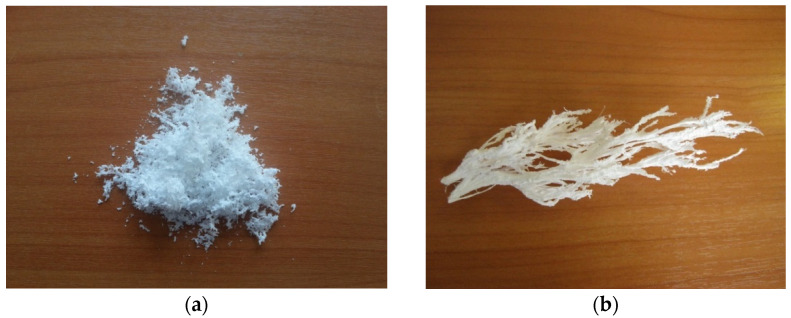
SAS formation of polycarbonate (PC) powder at 1 mL/min (**a**) and PC icicles at 2 mL/min (**b**).

**Figure 4 materials-14-07428-f004:**
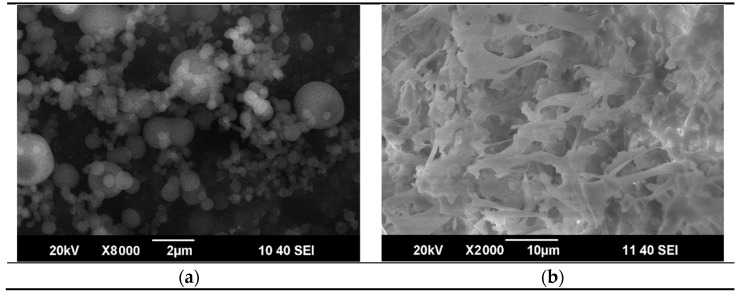
Scanning electron microscopy (SEM) images of the PC particles micronized at the 1 mL/min (**a**) and 2 mL/min (**b**) solution flow rates. 150 bar, 25 g/L.

**Figure 5 materials-14-07428-f005:**
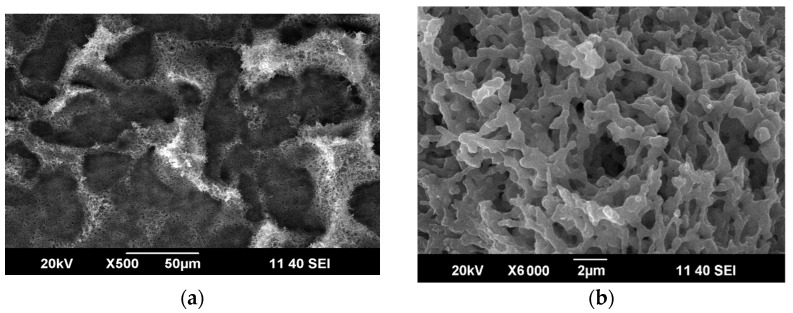
SEM images of the PC-carboxylated carbon nanotube (CNT) composite powder at 500× (**a**) and 6000× zoom (**b**).

**Figure 6 materials-14-07428-f006:**
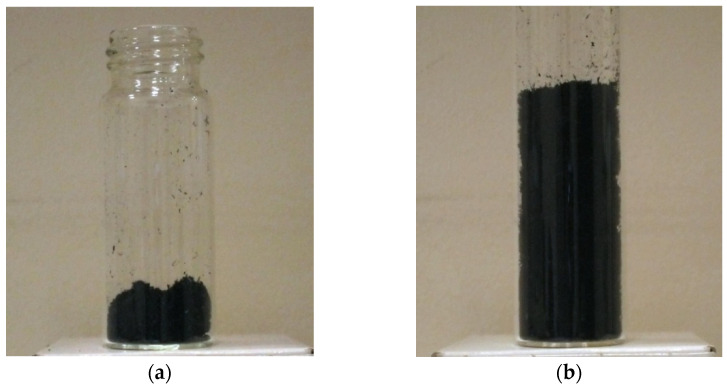
0.1 g of CNTs before (**a**) and after (**b**) RESS.

**Figure 7 materials-14-07428-f007:**
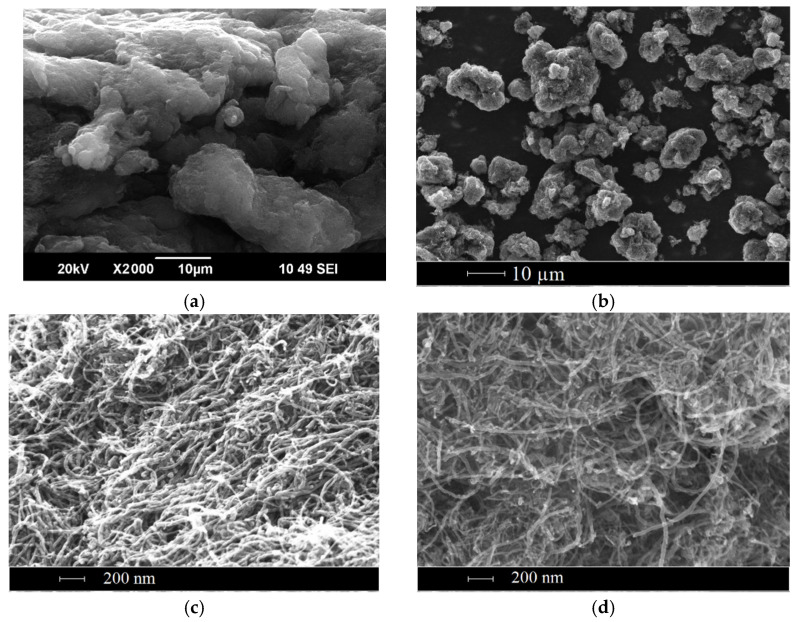
SEM images of initial CNTs (**a**,**c**) and CNTs (**b**,**d**) processed by RESS.

**Figure 8 materials-14-07428-f008:**
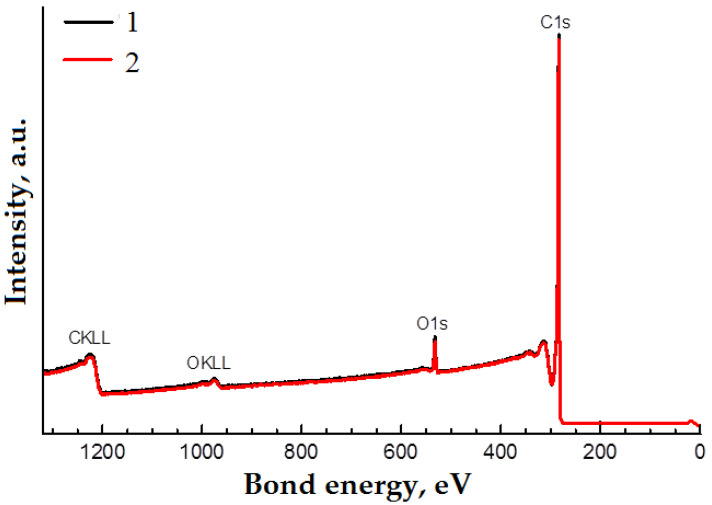
XPS spectra of initial CNTs (1) and CNTs after RESS (2).

**Figure 9 materials-14-07428-f009:**
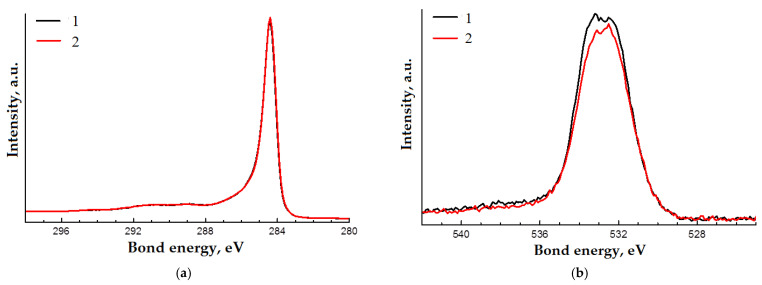
C1s (**a**) and O1s (**b**) XPS spectra of initial CNTs (1) and CNTs after RESS (2).

**Figure 10 materials-14-07428-f010:**
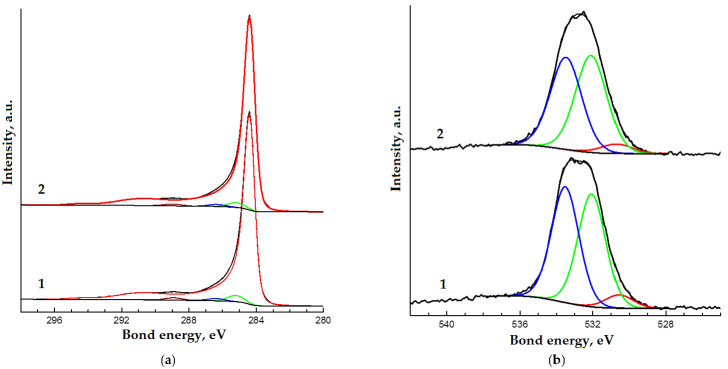
Decomposition into components C1s (**a**) and O1s (**b**) XPS spectra of initial CNTs (1) and CNTs after RESS (2).

**Figure 11 materials-14-07428-f011:**
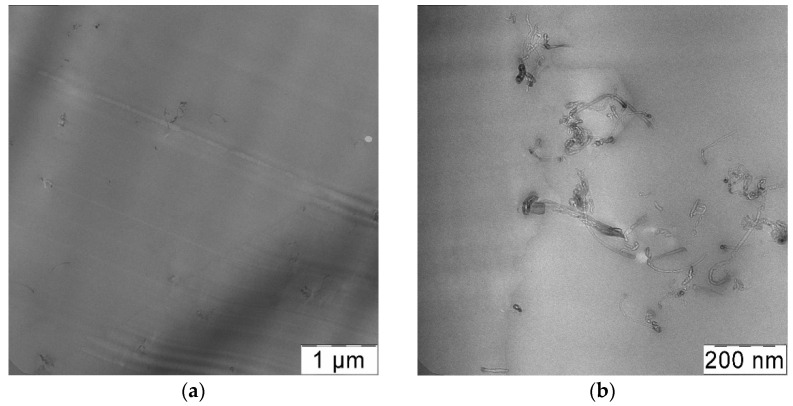
Transmission electron microscopy (TEM) images of composites obtained by SAS with untreated CNTs (**a**,**b**) (experiment No. 1) and CNTs processed by RESS (**c**,**d**) (experiment No. 3).

**Figure 12 materials-14-07428-f012:**
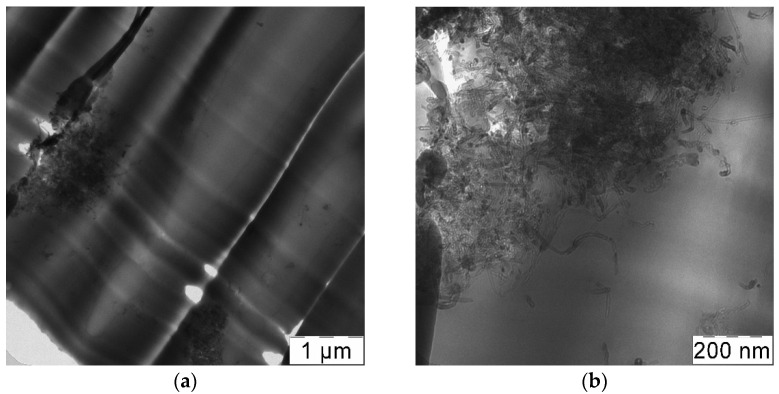
TEM images of composites obtained via solution processing with untreated CNTs (**a**,**b**) (experiment No. 2) and CNTs processed by RESS (**c**,**d**) (experiment No. 4).

**Figure 13 materials-14-07428-f013:**
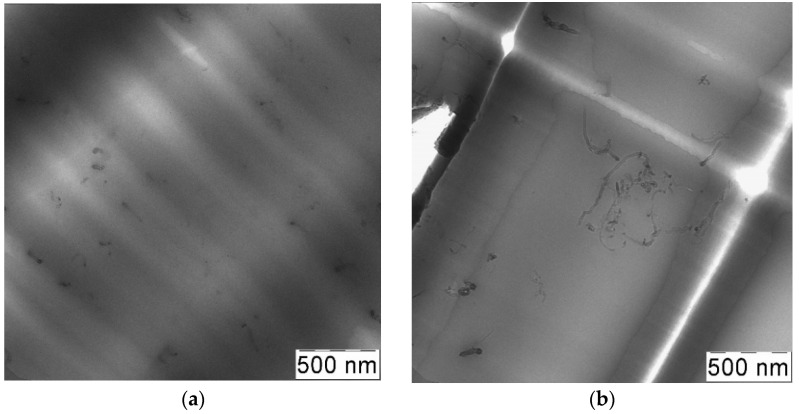
TEM images of composites obtained via SAS with powerful ultrasound treatment: untreated CNTs (**a**) (experiment No. 5) and CNTs processed by RESS (**b**) (experiment No. 6).

**Table 1 materials-14-07428-t001:** Element concentrations, bond energies and types of bonds in the X-ray photoelectron spectroscopy (XPS) spectra.

Spectrum	The Content of the Element, Atomic %		Bond Energy, eV	Atomic %		Type of Bond
	Initial CNTs	CNTs after RESS		Initial CNTs	CNTs after RESS	
O1s	3.2	3.0	530.6	0.2	0.1	O^−^
			532.1	1.5	1.5	O−C (aliphatic)
			533.5	1.5	1.4	O=C (aliphatic)
C1s	96.8	97.0	284.4	91.7	92.8	C−C (sp^2^)
			285.2	2.9	2.3	C−C,H (sp^3^)
			286.4	1.1	1.0	C−O
			288.9	1.1	0.9	O=C−O

**Table 2 materials-14-07428-t002:** Mechanical properties of PC-cylindrical carboxylated CNT composites obtained by different methods.

Exp. No.	Composite Preparation Technique	Young’s Modulus, GPa	Tensile Strength, MPa	Total Elongation, %	Elongation in the Elastic Range, %
0	Initial PC for comparison	1.48 ± 0.06	28 ± 2	1.3 ± 0.1	1.22 ± 0.07
1	SAS without additional pre-treatments	1.49 ± 0.01	48 ± 2	4 ± 1	1.73 ± 0.06
2	Solution processing without additional pre-treatment	1.53 ± 0.09	35 ± 4	2.0 ± 0.2	1.4 ± 0.1
3	SAS. The CNTs were pre-processed by RESS	1.53 ± 0.02	58 ± 1	5 ± 1	1.6 ± 0.1
4	Solution processing. The CNTs were pre-processed by RESS	1.38 ± 0.01	40 ± 3	2.6 ± 0.3	1.3 ± 0.1
5	SAS. Powerful ultrasound was used	1.50 ± 0.04	50 ± 5	8.3 ± 0.9	1.63 ± 0.08
6	SAS. The CNTs were pre-processed by RESS and powerful ultrasound was used	1.4 ± 0.1	45 ± 5	3.2 ± 0.7	1.6 ± 0.1

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
