# Peer review of "Formation of Polymer-Carbon Nanotube Composites by Two-Step Supercritical Fluid Treatment"

_materials, 2021, doi:10.3390/ma14237428_

Round 1

Reviewer 1 Report

An approaсh for polymer- carbon nanotube (CNT) composite preparation is proposed based on a two-step supercritical fluid treatment. The first step, rapid expansion of a suspension RESS) of CNTs in supercritical carbon dioxide, is used to de-bundle CNTs in order to simplify their mixing with polymer in solution. The ability of RESS pre-treatment to de-bundle CNTs and to cause significant bulk volume expansion is demonstrated. The second step is a formation of polymer-CNT composite from solution via supercritical antisolvent (SAS) precipitation. The combination of these two supercritical fluid methods allowed obtaining poly carbonate - multiwalled carbon nanotube composite with tensile strength two times higher compared to initial polymer and enhanced elasticity. Overall, the paper presents a nice expt approach on the use of CNTs and reports interesting findings. The referee feels the publication of this paper is within the scope of Materials. The authors need to address the following minor comments/issues before its consideration:

- Authors can elaborate the references instead of citing/combining in section 1.

-Research gap and motivation are nicely presented.  

-Synthesis of CNTs can be explained in section 2.1 as authors mentioned the procedure described elsewhere [48]. This would help readers.

- Section 2.3 Preparation of polymer-CNT solutions also needs to expand.

-Please add reason for the statement “However, the solution processing method does not allow achieving the results of the SAS method alone (exp. № 1) even if it uses RESS-processed CNTs …….”

- The following literature on the CNT-based nanocomposites using different techniques and ultrasonication needs to be discussed as authors talk about challenges of CNT-polymer composites formation: Polymer Composites 41(6), 2491-2507 (2020)

Author Response

Response to Reviewer 1 Comments

The authors thank the reviewer for his work and valuable comments. The authors agree with reviewer's remarks. Appropriate changes have been made:

- Authors have elaborated the references in section 1.

-Synthesis of CNTs is explained in section 2.1.

- Section 2.3 Preparation of polymer-CNT solutions is expanded.

-The reason for the statement “However, the solution processing method does not allow achieving the results of the SAS method alone (exp. № 1) even if it uses RESS-processed CNTs …….” is added.

- «Polymer Composites 41(6), 2491-2507 (2020)» is discussed.

Reviewer 2 Report

The work entitles "Formation of polymer - carbon nanotube composites by two- step supercritical fluid treatment" reports preparation of PC/CNT composites via different processing methods. I recommend a major revision for this manuscript.

Comments:

- Since the authors have synthesized the nanotubes and modified them, they need to report the SEM micrographs and XPS curves/data of the synthesized CNT and modified CNT.

- Figure. 5. SEM image. It should be SEM images.

- Figure 7: Again images not image.

- Mechanical properties:  The authors should discuss the role of RESS in improving distribution of the nanotubes and cite references from literature.

- Presence of the nanotube in PC matrix increases not only  strength but also elongation. Why? I think that is because of the intrinsic flexibility of nanotubes. Please explain this phenomenon and cite refs from literature. You can  see and cite these refs which report a same observation: a) Poly(vinyl chloride)/single wall carbon nanotubes composites: Investigation of mechanical and thermal characteristics  

- "The use of powerful ultrasound (exp. № 5) virtually does not change the composite tensile strength, but leads to a two-fold increase in the total elongation in comparison with a regular ultrasonic bath (exp. № 1)". Why? Please justify this important observation.

Author Response

Response to Reviewer 2 Comments

The authors thank the reviewer for his work and valuable comments. The authors agree with reviewer's remarks. Appropriate changes have been made:

- The SEM micrographs and XPS curves/data of the synthesized and modified CNT are reported. XPS is also added to Materials and methods.

- Figure. 5, 7. captions are changed. Moreover, new images of CNT 7c and 7d at another zoom are added.

- The role of RESS in improving distribution of the nanotubes is discussed.

- "Presence of the nanotube in PC matrix increases not only strength but also elongation." The explanation is added.

- "The use of powerful ultrasound (exp. № 5) virtually does not change the composite tensile strength, but leads to a two-fold increase in the total elongation in comparison with a regular ultrasonic bath (exp. № 1)". The explanation is added.

Round 2

Reviewer 2 Report

This version of the manuscript entitled "Formation of polymer - carbon nanotube composites by two- step supercritical fluid treatment" have been revised according to the reviewers' comments. . So I recommend publication of this manuscript in materials.